# Semiparametric Causal Sufficient Dimension Reduction of Multidimensional Treatments

**Razieh Nabi**[1]  **Todd McNutt**[2]  **Ilya Shpitser**[3]

[1]Department of Biostatistics and Bioinformatics, Emory University, Atlanta, Georgia, USA
[2]School of Medicine, Johns Hopkins University, Baltimore, Maryland, USA
[3]Department of Computer Science, Johns Hopkins University, Baltimore, Maryland, USA

## Abstract

Cause-effect relationships are typically evaluated by comparing outcome responses to binary treatment values, representing two arms of a hypothetical randomized controlled trial. However, in certain applications, treatments of interest are continuous and multidimensional. For example, understanding the causal relationship between severity of radiation therapy, summarized by a multidimensional vector of radiation exposure values and post-treatment side effects is a problem of clinical interest in radiation oncology. An appropriate strategy for making interpretable causal conclusions is to reduce the dimension of treatment. If individual elements of a multidimensional treatment vector weakly affect the outcome, but the overall relationship between treatment and outcome is strong, careless approaches to dimension reduction may not preserve this relationship. Further, methods developed for regression problems do not directly transfer to causal inference due to confounding complications. In this paper, we use semiparametric inference theory for structural models to give a general approach to causal sufficient dimension reduction of a multidimensional treatment such that the cause-effect relationship between treatment and outcome is preserved. We illustrate the utility of our proposals through simulations and a real data application in radiation oncology.

## 1 INTRODUCTION

In causal inference, the exposure of interest is commonly assumed to be either binary (e.g., comparing treatment vs placebo) or continuous (e.g., effect of treatment dosages on viral load.) In the latter cases, in addition to contrasts of responses to two specific doses, we may be interested in the entire dose-response relationship, and choose to model it via a simple functional, for example a logarithmic or sigmoidal function. In other applications, we might be interested in assessing causal relationships between outcomes and treatments with values that lie in a multidimensional space. For instance, in natural language processing interest lies in causal analyses that involve high dimensional text data [Gentzkow et al., 2019, Feder et al., 2021]. Another example is the neuroimaging data used to relate neuronal network activity to cognitive processing and behavior [Ramsey et al., 2010, Mather et al., 2013].

As our motivational example, we focus on an application in radiation oncology. In neck and head cancers, minor variations in dose and direction of radiation may result in similar tumor reduction but vastly improve secondary outcomes, such as weight loss, or dysfunction induced by radiation therapy, such as dysphasia or xerostomia [Robertson et al., 2015]. Thus, understanding the causal relationship between a multidimensional radiation exposure and downstream side effects in cancer patients undergoing radiation therapy is of clinical interest. Unlike standard treatments, radiation therapy is complex and is represented by three dimensional voxel maps of radiation doses in different parts of the body. Since this representation is very high dimensional, the exact dose localization information in the voxel map is sometimes represented by cumulative dose-volume histograms and summarized by a multidimensional vector of exposure dosages. Even such summaries complicates establishing clinically relevant causal relationships.

Since we are interested in dimension reduction for the sake of explicating a particular relationship between treatments and outcomes, approaches that do not take outcomes into account in the right way run the risk of distorting the estimate of this relationship, or even falsely concluding the relationship is absent. Therefore, seemingly natural approaches to dimension reduction, such as principal component analysis (PCA), are not appropriate in our setting. On the other hand, there is a line of research in statistics on sufficient dimension reduction (SDR) [Li, 1991] with the objective of reducing

*Accepted for the 38th Conference on Uncertainty in Artificial Intelligence* (UAI 2022).

the dimension of covariates by preserving the associational relations between covariates and outcome. However, due to spurious associations introduced by confounding which is ubiquitous in observational data sources, naive use of SDR approaches to discern causal relationships between treatments and outcomes leads to bias.

We are interested in applying SDR core ideas to reduce dimension of a treatment in a way that preserves a *causal* rather than *associational* relationship with the outcome. In addition, we are interested in doing so under the weakest possible assumptions, which entails generalizing the semiparametric approaches in the SDR literature [Ma and Zhu, 2012]. In this paper, we provide a framework for structural (causal) models based on semiparametric inference theory developed for marginal structural models [Robins, 1999] to give what we believe is the first approach to causal SDR of a multidimensional treatment.

## 2 PRELIMINARIES

**Sufficient dimension reduction**. Given an outcome variable $Y$ and a $p$-dimensional covariate vector $X$, the goal of SDR is to find a known function $g_X(.; \beta)$ parameterized by $\beta$ with a much smaller range than domain such that $Y$ depends on $X$ only through $g_X(X; \beta)$. Often this function is assumed to be linear, in which case the goal is to find $\beta \in \mathbb{R}^{p \times d}$, where $d < p$, such that $Y$ depends on $X$ only through $X^T \beta$, i.e., $\mathbb{E}[Y|X] = \mathbb{E}[Y|X^T\beta]$. Often, proposed solutions to SDR rely on strong parametric assumptions that are unlikely to hold in practical applications, such as the linearity condition where $\mathbb{E}[X|X^T\beta]$ is assumed to be a linear function of $X$, or the assumption that $\text{cov}(X \mid X^T\beta)$ is constant rather than a function of $X$, [Li, 1991, Cook and Weisberg, 1991, Hardle and Stoker, 1989, Ichimura, 1993, Cook and Li, 2002].

Ma and Zhu [2012] introduced a new approach to SDR by recasting the problem in terms of estimation in a semiparametric model. Crucially, this approach relies on far weaker assumptions than is typical in SDR, and is thus much more generally applicable. To obtain the relevant semiparametric model, we rewrite the above condition as $Y = \ell(X^T\beta) + \epsilon$, where $\ell(X^T\beta) := \mathbb{E}[Y|X^T\beta]$ is an unspecified smooth function and $\mathbb{E}[\epsilon|X] = 0$, while the distribution $p(\epsilon|X)$ remains otherwise unrestricted. Ma and Zhu [2012] derived the class of all influence functions for $\beta$, a.k.a. the orthogonal nuisance tangent space denoted by $\Lambda_\eta^\perp$, as: $\Lambda_\eta^\perp = \{(Y - \mathbb{E}[Y|X^T\beta]) \times (\alpha(X) - \mathbb{E}[\alpha(X)|X^T\beta])\}$, where $\alpha(X)$ is any function of $X$.

The main objective of semiparametric theory of influence functions is to derive an estimator that has the characteristics of parametric models (such as a $\sqrt{n}$-consistency and asymptotic normality) without the restrictive parametric assumptions. Influence functions are arguably the most important approach to estimation in causal inference, especially in

observational studies where we do not have prior knowledge on what the true likelihood is; see Appendix A for a brief overview of influence functions, and [Van der Vaart, 2000, Bang and Robins, 2005, Tsiatis, 2007] for more details.

A well-known property of semiparametric models is that all elements of $\Lambda_\eta^\perp$ are mean 0 under the true distribution. Hence, a general class of estimating equations can be obtained using the sample version of

$$\mathbb{E}[U(\beta)] \hspace{3cm} (1)$$
$$= \mathbb{E}\Big[(Y - \mathbb{E}[Y \mid X^T\beta]) \times (\alpha(X) - \mathbb{E}[\alpha(X) \mid X^T\beta])\Big] = 0,$$

where $U(\beta)$ is an arbitrary element in $\Lambda_\eta^\perp$. The estimator obtained from (1) is doubly robust under any choice of models for $\mathbb{E}[Y|X^T\beta]$ and $\mathbb{E}[\alpha(X)|X^T\beta]$, meaning that the estimator remains consistent if either of these two models is correctly specified [Ma and Zhu, 2012].

**Causal inference.** In causal inference, we seek to make inferences about the causal relationship of a treatment variable $A$ and an outcome variable $Y$ by counterfactual contrasts of the form $Y(a)$ representing a hypothetical experiment where treatment $A$ is set to $a$, possibly contrary to the fact. A common setting considers, in addition to $A$ and $Y$, a vector of baseline variables $C$, yielding an observed data distribution of the form $p(Y, A, C)$. Under standard assumptions of *consistency*, which states that counterfactual outcome is the same as observed outcome if treatment is set to observed value, *conditional ignorability* which states that $\{Y(a)\}$ is independent of $A$ conditional on $C$, and *positivity* of $p(A \mid C)$, the counterfactual distribution $p(Y(a))$ is identified as the following function of observed data

$$p(Y(a)) = \sum_c p(Y \mid A = a, C = c) \times p(C = c). \quad (2)$$

The *average causal effect* (ACE) of a binary treatment on an outcome is defined as $\text{ACE} = \mathbb{E}[Y(1)] - \mathbb{E}[Y(0)]$. Under the above assumptions, the counterfactual mean $\mathbb{E}[Y(a)]$ is given as the following function of the observed data, called the *adjustment formula* or *g-formula*,

$$\mathbb{E}[Y(a)] = \mathbb{E}\big[\mathbb{E}[Y \mid A = a, C]\big], \quad (3)$$

where the outer expectation is taken with respect to $p(C)$; see Tian and Pearl [2002], Shpitser and Pearl [2006], Bhattacharya et al. [2020] for general identification algorithms in the presence of unmeasured confounders.

There are several different approaches on estimating the adjustment formula such as plug-in, inverse probability weighting (IPW), and semiparametric based estimators such as augmented IPW (AIPW) [Van der Vaart, 2000, Bang and Robins, 2005, Van der Laan et al., 2011]. An alternative class of IPW estimators models the relationship between $A$ and $Y$ via a *marginal structural model (MSM)*, or a causal regression. A simple version of such a model takes the form $\mathbb{E}[Y(a)] = f(a; \beta)$, for finite set of parameters $\beta$. Given

such a model, inferences about $\mathbb{E}[Y(a)]$ reduce to inferences about $\beta$. For binary treatments, $f(a; \beta)$ can be written as $\beta_0 + \beta_a \times a$ without loss of generality, with $ACE = \beta_a$. An MSM is different from an ordinary regression model, since $\mathbb{E}[Y(a)] \neq \mathbb{E}[Y|A = a]$ given our causal assumptions. Thus, one approach to estimating $\beta$ is via the following estimating equation, appropriately reweighted by the treatment propensity score model, $W_a(C; \eta_a) := p(A|C; \eta_a)$,

$$\mathbb{P}_n \left[ \frac{p^*(a)}{W_a(C; \widehat{\eta}_a)} \times \{Y - f(a; \beta)\} \right] = 0, \qquad (4)$$

where $\mathbb{P}_n = \frac{1}{n} \sum_{i=1}^{n} (.)$, $p^*(a)$ is an arbitrary function of $a$ with the same dimension as $\beta$, and $\widehat{\eta}_a$ is the maximum likelihood estimate of $\eta_a$. This IPW procedure is known to be inefficient. A more efficient (in fact optimal in a wide class of reasonable estimators) approach is to use influence functions, described in detail in [Robins, 1999]. Our approach to causal SDR stands in the same relation to the semiparametric approach to SDR for regression problems in [Ma and Zhu, 2012] as fitting regression models does to fitting marginal structural models.

## 3 CAUSAL SUFFICIENT DIMENSION REDUCTION

We are interested in the causal effect of a multidimensional treatment $A \in \mathbb{R}^p$ on outcome $Y$, assuming all relevant covariates that need to be controlled for are observed and denoted by $C$. We would like to reduce the dimension of treatment $A$ such that the causal relationship between $A$ and $Y$ is preserved. Let $g(.; \beta)$ be a function parameterized by $\beta$ that takes values in $\mathbb{R}^p$ and maps them to values in $\mathbb{R}^d$, $d < p$, i.e., $g : A \in \mathbb{R}^p \mapsto g(A; \beta) \in \mathbb{R}^d$. We want to reduce the dimension of $A$ in such a way that the counterfactual response $\mathbb{E}[Y(a)]$ only depends on $A$ via $g(a)$. Specifically, we assume that if $\mathbb{E}[Y(a)]$ is identified, that is if $\mathbb{E}[Y(a)]$ is a mapping $f$ from values $a$ of $A$ to functionals $h_a(p(V))$ of the observed data distribution, where $p(V)$ denotes the joint distribution over the set of observed variables $V$, then $f(a) = f(g(a; \beta))$. The methodology proposed in this paper does not depend on the choice of $g(.; \beta)$, although we fix a particular $g(.; \beta)$ in our experiments. We assume the three identification assumptions that were discussed in the previous section, namely consistency, conditional ignorability, and positivity, hold in our analysis. Therefore, we fix $h_a(p(C, A, Y)) = \mathbb{E}[\mathbb{E}[Y|A = a, C]]$, as shown in (2).

The estimation procedure for MSMs shown in (4) can be viewed as a standard estimating equation for a regression model relating treatment and outcome, but applied to observed data readjusted via inverse weighting in such a way that treatment appear randomly assigned. In other words, MSMs are regressions applied to a version of observed data in such a way that regression parameters can be interpreted causally. Unlike other estimating equations that

solve for $\beta$ by maximizing the feature outcome relationship, the equation in (1) fits $\beta$ to maintain the identity $\mathbb{E}[Y|X] = \mathbb{E}[Y|X^T \beta]$. As a consequence, semiparametric causal SDR can be viewed as an MSM version of this regression problem, which seeks to find $\beta$ which maintains $\mathbb{E}[Y(a)] = \mathbb{E}[Y(g(a; \beta))]$. In other words, our aim is to estimate $\beta$ by maintaining the following identity

$$\mathbb{E}\Big[\mathbb{E}\big[Y \mid a, C\big]\Big] = \mathbb{E}\Big[\mathbb{E}\big[Y \mid g(a; \beta), C\big]\Big], \qquad (5)$$

where the outer expectation is wrt the density $p(C)$.

We note here the different roles that variables play in regression SDR and causal SDR. The goal of regression SDR is to preserve the associative relationship between high dimensional features $X$ and outcome $Y$. The goal of causal SDR, as we view it here, is to preserve the causal relationship between a multidimensional treatment $A$ and outcome $Y$, which is made complicated by the presence of spurious associations induced by covariates $C$. Thus, the goal in causal SDR is *not* to maintain the regression relationship between covariates and outcome by assuming $\mathbb{E}[Y|\{A, C\}] = \mathbb{E}[Y|g(\{A, C\}; \beta)]$, but to preserve the relationship as in (5) where $C$ is marginalized (adjusted for). The set of confounders $C$ could still be high dimensional, but they are not of primary interest in our problem. Incorporating baseline covariates into the dimension reduction strategy along with the treatment, as is done in some MSMs, is left as an interesting avenue for future work. Examples of work focusing on dimension reduction of common confounders include Imai and Ratkovic [2014], Hu et al. [2014], Shortreed and Ertefaie [2017], Banijamali et al. [2018], Ma et al. [2019], Luo and Zhu [2020], Cheng et al. [2020].

As stated earlier, our objective is to preserve the causal effect of $A$ on $Y$, which is of the form shown in (3). However, it suffices to say that if the counterfactual response curve, i.e., $\mathbb{E}[Y(a)]$, is preserved under our dimensionality reduction scheme, then the causal effect is preserved. Hence, we stated our constraint in (5) in terms of the counterfactual mean rather than the counterfactual contrast that would define the effect. Moreover, even though treatment is multidimensional, we emphasize that each unit still receives one treatment session; e.g., a single session of radiation therapy with no followups. Records of radiation treatment are usually stored as monodimentional cumulative dose-volume histograms, and are summarized as amount of radiation on $k\%$ of the organ's volume, where $k$ ranges from 1 to 100.

In a conditionally ignorable causal model, intervention on $A$ corresponds to dropping the term $p(A|C)$ from the observed density $p(Y, A, C)$ yielding (2). Define $q(Y, A, C)$ as the following modified version of (2):

$$q(Y, A, C) := p(Y \mid A, C) \times p^*(A) \times p(C),$$

where $p^*(A)$ is any density with the same support as $p(A)$.

Then (5) can be rewritten as

$$\mathbb{E}_q[Y \mid A = a] = \mathbb{E}_q[Y \mid g(a; \beta)], \qquad (6)$$

where $\mathbb{E}_q$ is the expectation taken with respect to the density $q(Y, A, C)$ defined above, and $q(Y|A) = \sum_C q(Y, C|A) = \sum_C p(Y|A, C) \times p(C)$ by definition.

Equations (5) and (6) are equivalent forms of our constraint in the causal SDR problem where the MSM model for $\mathbb{E}[Y(a)] = \mathbb{E}_q[Y|a]$, is assumed to be a function of the multidimensional treatment intervention $a$ only through its lower dimension representation $g(a; \beta)$. We now describe two approaches to estimating $\beta$.

## 3.1 INVERSE PROBABILITY WEIGHTED SDR

Let $\ell(g(A; \beta)) := \mathbb{E}_q[Y|g(A; \beta)]$ and $\nu(g(A; \beta)) := \mathbb{E}_q[\alpha(A) \mid g(A; \beta)]$ be two unspecified smooth functions of $g(A; \beta)$. A simple estimation strategy for $\beta$ based on generalizing (4), entails solving

$$\mathbb{E}\left[\frac{p^*(a)}{p(A = a \mid C)} \times \widetilde{U}(\beta)\right] = 0, \qquad (7)$$

where $\widetilde{U}(\beta) = \{Y - \ell(g(a; \beta))\} \times \{\alpha(A) - \nu(g(a; \beta))\}$, $p^*(a)$ is an arbitrary function of $a$, and $p(A|C)$ is a correctly specified statistical model which governs how the treatment $A$ is assigned based on baseline characteristics $C$. The above equation may be solved using observed data by evaluating the expectation empirically.

**Lemma 1.** *An estimator for $\beta$ based on solving (7) is unbiased under correct specification of $p(A|C)$, and either one of $\ell(g(A; \beta)) := \mathbb{E}_q[Y|g(A; \beta)]$ or $\nu(g(A; \beta)) := \mathbb{E}_q[\alpha(A) \mid g(A; \beta)]$.*

## 3.2 SEMIPARAMETRIC CAUSAL SDR

A general approach for deriving *regular and asymptotically linear* (RAL) estimators of $\beta$ is based on deriving $\widetilde{\Lambda}_\eta^\perp$, the orthogonal complement of the nuisance tangent space of a semiparametric model that enforces the constraint (5), but places no other restrictions on the observed data distribution; $\widetilde{\Lambda}_\eta^\perp$ is the class of all influence functions. One approach is to derive this space explicitly, as was done in [Ma and Zhu, 2012]. An alternative is to take advantage of general theory relating orthogonal complements of regression problems, and orthogonal complements of "causal regression problems," or MSMs, developed by Robins [1999]. Given the semiparametric model $\mathcal{M}$ induced by the restriction (5), we take advantage of this theory in the following result.

**Theorem 1.** *The orthogonal complement of the nuisance tangent space $\widetilde{\Lambda}_\eta^\perp$ for $\beta$ that satisfies (6) is:*

$$\widetilde{\Lambda}_\eta^\perp = \left\{\frac{\widetilde{U}(\beta)}{W_a(C)} - \phi(A, C) + \mathbb{E}[\phi(A, C) \mid C]\right\},$$

*where $\phi(A, C)$ is an arbitrary function of $A$ and $C$, $W_a(C)$ is the IPW weight $p(A = a|C)/p^*(a)$ for a fixed $p^*(a)$, and $\widetilde{U}(\beta)$ is of the form*

$$\widetilde{U}(\beta) = \{Y - \ell(g(a; \beta))\} \times \{\alpha(A) - \nu(g(a; \beta))\},$$

*where $\ell(g(a; \beta)) := \mathbb{E}_q[Y|g(a; \beta)]$ and $\nu(g(a; \beta)) := \mathbb{E}_q[\alpha(A)|g(A; \beta)]$. Moreover, the most efficient estimator in this class, for any fixed $\alpha(A)$, is recovered by setting $\phi^{opt}(A, C) = \mathbb{E}\left[\frac{\widetilde{U}(\beta)}{W_a(C)} \mid A, C\right]$.*

**Lemma 2.** *For a fixed choice of $\alpha(A)$ and normalized function $p^*(A)$, the element $\widetilde{U}(\beta^*) \in \widetilde{\Lambda}_\eta^\perp$ corresponding to the optimal choice of $\phi(A, C)$ has the form.*

$$\frac{p^*(A)}{p(A \mid C)} \times \widetilde{U}(\beta) - \frac{p^*(A)}{p(A \mid C)} \times \mathbb{E}\left[\widetilde{U}(\beta)\big|A, C\right]$$
$$+ \mathbb{E}_q\left[\mathbb{E}\left[\widetilde{U}(\beta)\big|A, C\right]\big|C\right], \qquad (8)$$

*where $\mathbb{E}_q[.]$ is the expectation taken with respect to the density $q(Y, A, C) := p(Y|A, C) \times p^*(A) \times p(C)$.*

## 3.3 ROBUSTNESS PROPERTIES

Just as $\Lambda_\eta^\perp$ in Section 2 entailed double robustness of $U(\beta)$ for semiparametric regression SDR, we now show that the structure of $\widetilde{\Lambda}_\eta^\perp$ yields additional robustness properties.

**Lemma 3.** *If one of $\{p(A|C), \mathbb{E}[\widetilde{U}(\beta)|A, C]\}$ and one of $\{\ell(g(A; \beta)) := \mathbb{E}_q[Y|g(A; \beta)], \nu(g(A; \beta)) := \mathbb{E}_q[\alpha(A)|g(A; \beta)]\}$ is correctly specified, then the estimator for $\beta$ based on (8) is consistent and asymptotically normal with mean zero and variance $\tau^{-1} \times Var(\widetilde{U}(\beta^*)) \times \tau^{-1'}$, where $\widetilde{U}(\beta^*)$ is given in (8) and $\tau = \mathbb{E}[\partial\widetilde{U}(\beta^*)/\partial\beta]$.*

This result implies that the estimating equation in (8) yields a "2 × 2" robustness property. In practice, since we will be dealing with multidimensional problems, correct specification of models is difficult to ensure. However, robustness properties of semiparametric estimators also implies that in regions where sufficient subset of models are approximately correct, the overall bias remains small. If $p(A|C)$ and one of the models in $\widetilde{U}(\beta)$ is correctly specified, the AIPW estimator using (8) remains consistent for any choice of $\mathbb{E}[\widetilde{U}(\beta)|A, C]$. One promising direction of future work is to consider cases where $p(A|C)$ and $\widetilde{U}(\beta)$ are known and search for $\mathbb{E}[\widetilde{U}(\beta)|A, C]$ which yields good properties of the overall estimator.

# 4 ESTIMATION AND IMPLEMENTATION

In order to estimate the parameters $\beta$ in 6, we need to solve the estimating equation $\mathbb{P}_n[\widetilde{U}(\beta^*)] = 0$, where $\widetilde{U}(\beta^*)$ is given in (8). For any $\widetilde{U}(\beta)$ of the form given in Section

3.1, Theorem 1, provides the class of all RAL estimators for $\beta^*$ along with the most efficient estimator in this class. Under the general form of $\widetilde{U}(\beta) = \{Y - \ell(g(A; \beta))\} \times \{\alpha(A) - \nu(g(A; \beta))\}$, the term $\mathbb{E}[\widetilde{U}(\beta)|A, C]$ in $\widetilde{U}(\beta^*)$ equals $\{\mathbb{E}[Y|A, C] - \ell(g(A; \beta))\} \times \{\alpha(A) - \nu(g(A; \beta))\}$. Hence, in the expression in 8, four different models are involved in estimating $\widetilde{U}(\beta^*)$, namely (i) $\ell(g(A; \beta)) := \mathbb{E}_q[Y|g(A; \beta)]$, (ii) $\nu(g(A; \beta)) := \mathbb{E}_q[\alpha(A)|g(A; \beta)]$, (iii) $p(A|C)$, and (iv) $\mathbb{E}[Y|A, C] = \mathbb{E}_q[Y|A, C]$. The last term in (8) is equal to $\mathbb{E}_a[\mathbb{E}[U(\beta)|A, C]]$, where $\mathbb{E}_a[.]$ is the expectation wrt the marginal distribution of $A$ which can be evaluated empirically without additional modeling.

For a pre-specified functional form of $\ell(g(A; \beta))$, we need to fit three different nuisance models. Given models $\nu(g(A; \beta); \eta_\nu)$, $p(A|C; \eta_a)$, and $\mathbb{E}[Y|A, C; \eta_y]$ for $\nu(g(A; \beta))$, $p(A|C)$, and $\mathbb{E}[Y|A, C]$, respectively, it can be shown that if $n^{\frac{1}{4}+\epsilon}(\widehat{\eta} - \eta_0)$ is bounded in probability for some $\epsilon > 0$, then the estimating equation $\mathbb{P}_n[\widetilde{U}(\beta^*); \widehat{\eta}] = 0$ yields an estimate of $\beta$ with the same asymptotic properties as if the nuisance models were known. Here $\eta = \{\eta_\nu, \eta_a, \eta_y\}$, and $\widehat{\eta}, \eta_0$ denote the estimated and the true parameters of the nuisance models, respectively.

**Theorem 2.** *Let $\phi_0$ denote the influence function of the estimator $\beta$ obtained from the estimating equation $\mathbb{P}_n[\widetilde{U}(\beta^*, \eta_0)] = 0$. If $n^{\frac{1}{4}+\epsilon}(\widehat{\eta} - \eta_0)$ is bounded in probability for some $\epsilon > 0$, then the influence function corresponding to the estimator $\widehat{\beta}$ obtained from the estimating equation $\mathbb{P}_n[\widetilde{U}(\beta^*, \widehat{\eta})] = 0$ is the same as $\phi_0$. In other words, $\widehat{\beta}$ follows the same asymptotic properties as if we knew the true nuisance models.*

The condition for the rate of convergence of nuisance models in Theorem 2 is a sufficient condition and is potentially too conservative. In practice, we might be able to use models with the slower convergence rates, see [Fisher and Kennedy, 2018] for more details. [Stone, 1982] provides a detailed analysis of the convergence rates of nonparametric models.

**Implementation**. We now describe in detail our procedure for estimating $\beta$ by solving the empirical version of $\mathbb{E}[\widetilde{U}(\beta^*)] = 0$, where $\widetilde{U}(\beta^*)$ is given in (8). In what follows, we assume the structural dimension $d$, i.e. the cardinality of the range of $g(; \beta)$, is known; we provide a discussion on choosing the structural dimension at the end of this section.

For a given choice of $p^*(A)$ and $\alpha(A)$,

1. First estimate $\widehat{\eta}_a$ and $\widehat{\eta}_y$ in $p(A|C; \eta_a)$ and $\mathbb{E}[Y|A, C; \eta_y]$ by maximum likelihood or nonparametric methods. These two models do not depend on $\beta$ and are not updated within the iterations below.

2. Pick starting values $\beta^{(1)}$.

3. At $j^{\text{th}}$ iteration, given a fixed $\beta^{(j)}$, estimate $\widehat{\ell}(g(A; \beta^{(j)}))$ and $\widehat{\nu}(g(A; \beta^{(j)}))$, and compute:

$$U^q(\beta^{(j)}) = \{Y - \widehat{\ell}(g(A; \beta^{(j)}))\} \times \{\alpha(A) - \widehat{\nu}(g(A; \beta^{(j)}))\},$$
$$\mathbb{E}[U^q(\beta^{(j)}) \mid A, C] = \{\mathbb{E}[Y \mid A, C; \widehat{\eta}_y] - \widehat{\ell}(g(A; \beta^{(j)}))\}$$
$$\times \{\alpha(A) - \widehat{\nu}(g(A; \beta^{(j)}))\}.$$

4. Form the sample version of $\mathbb{E}[\widetilde{U}(\beta^*)]$ as follows.

$$\zeta(\beta^{(j)}) = \mathbb{P}_n\left[\frac{p^*(A)}{p(A \mid C; \widehat{\eta}_a)} \times \left\{U^q(\beta^{(j)}) - \right.\right.$$
$$\left.\left. - \mathbb{E}[U^q(\beta^{(j)}) \mid A, C]\right\} + \mathbb{E}_q\left[\mathbb{E}[U^q(\beta^{(j)}) \mid A, C] \mid C\right]\right]$$

where $\mathbb{P}_n[.] := \frac{1}{n} \sum_{i=1}^{n} [.]_i$.

5. Calculate the first and second derivatives of $\partial\{\|\zeta(\beta)\|^2\}/\partial\{\beta\}$ numerically and evaluate them at $\beta^{(j)}$, and use the Newton-Raphson update rule to update $\beta^{(j)}$.

6. Repeat steps (3) through (5) until convergence.

The implementation of an empirical evaluation of (7) follows a similar set of steps, except all steps pertaining to second and third terms of (8) are skipped. Moreover, in step 3 of the above implementation, we need to specify individual models for $\ell(g(A; \beta)) := \mathbb{E}_q[Y|g(A; \beta)]$ and $\mathbb{E}[Y|A, C] := \mathbb{E}_q[Y|A, C]$. However, due to variation dependence of these models, it may be difficult to fit these two models in a congenial way in general. We provide an alternative approach in the following section.

In order to deal with the issue of congeniality, we may opt to specify $\mathbb{E}_q[Y|g(A; \beta)]$ and $\widetilde{f}(A, C, \beta) = \mathbb{E}_q[Y|A, C] - \mathbb{E}_q[Y|g(A; \beta)]$, which yield a variationally independent specification of $\mathbb{E}_q[Y|g(A; \beta)]$ and $\mathbb{E}_q[Y|A, C] = \mathbb{E}_q[Y|g(A; \beta)] + \widetilde{f}(A, C, \beta)$. Consequently, the four variationally independent models we need to specify are as follows: $\ell(g(A; \beta))$, $\nu(g(A; \beta))$, $p(A|C)$, and $\widetilde{f}(A, C, \beta)$; the last term in (8) can be evaluated empirically without additional modeling. Thus, we need to specify the additional nuisance model $\widetilde{f}$. We propose to fit $\widetilde{f}$ by borrowing ideas from the theory of structural nested mean models (SNMMs) in [Vansteelandt and Joffe, 2014, Robins, 1999]. We defer the descriptions to the appendix and refer to $\widetilde{f}$ as an "inverted" structural nested mean model.

**Choosing the structural dimension**. Up until here, we assumed the structural dimension was known a priori. Finding the correct dimension is not an straightforward task and incorrect choices may greatly affect performance. We adapt the technique in [Ma and Zhu, 2012] that was used to select the structural dimension in regression SDR to causal SDR. Specifically, we utilize a resampling procedure to select the structural dimension. This procedure was originally described by [Dong and Li, 2010] and adapts the

idea of [Ye and Weiss, 2003]. We consider a family of functions $g^1(.; \beta^1), \ldots, g^m(.; \beta^m)$ with different structural dimensions, and use the cross-validation procedure we describe below to pick the best dimension.

Let $\widehat{\beta}_\rho$ be the estimate of $\beta$ from the original sample for the $\rho^{\text{th}}$ working dimension, where $\rho = 1, \ldots, p-1$, and let $\widehat{\beta}_{\rho,b}$ be the estimate of $\beta$ from the $b^{\text{th}}$ bootstrap sample, for $b = 1, \ldots, B$. The structural dimension can be estimated by finding the dimension $\rho$ to be the cardinality of the range of the function

$$g^* = \arg\max_{g^i} \frac{1}{B} \sum_{b=1}^{B} r^2\big(g^i(A; \widehat{\beta}_\rho), g^i(A; \widehat{\beta}_{\rho,b})\big),$$

where $r^2(u,v) = k^{-1} \sum_{i=1}^{k} \lambda_i$ and $\lambda_i$s are the non-zero eigenvalues of

$$\{\text{var}(u,v)\}^{-1/2} \text{cov}(u,v) \{\text{var}(v)\}^{-1} \text{cov}(v,u) \{\text{var}(u)\}^{-1/2}.$$

This procedure uses resampling to choose $\beta$ to maximize variability of the reduced set of features given by $g^i(.; \beta^i)$ where $g^i(.; \beta^i)$ is chosen in a way that aims to preserve the causal regression relationship between $A$ and the mean of $Y$. Exploring other alternatives for choosing the structural dimension is an interesting area for future work.

In the next section, we describe a set of simulations to illustrate the key results presented in this paper and provide a real-data analysis using a cohort of patients with head and neck cancer treated with radiation therapy. All code necessary to reproduce the results is available at https://github.com/raziehna/multidimensional-treatments.

## 5 SIMULATION STUDY

Causal SDR is not well-solved via standard methods for dimension reduction such as PCA, as they do not take the feature-outcome relationship into account, nor by standard SDR methods, as they do not take the confounding issues into account. We illustrate the utility of our proposal to causal SDR, via simulation studies, and compare them with regression SDR and PCA methods. We also illustrate the consistency of our estimators and illustrate the procedure for selecting the structural dimension. To provide continuity with previous work, our simulation study is similar to that described in [Ma and Zhu, 2012].

We perform 50 replications with fixed sample sizes, where the true response $\mathbb{E}[Y(g(a))]$ is an object of dimension $d = 2$, and the observed data distribution $p(Y, A, C)$ is set as follows. The dimension of the baseline factors $C$ is fixed as 4 and the observed treatment dimension $p$ is set to be 6 and 12. The baseline factors $C$ are generated from a standard multivariate normal distribution. We consider two cases for the treatment vector: one where the linearity and the constant covariance conditions in regular SDR are violated, and one where these assumptions are satisfied.

**Case 1.** We generated $(A_1, A_2)^T$ (when $p = 6$) and $(A_1, A_2, A_{7:12})^T$ (when $p = 12$) from a multivariate normal distribution where the mean of each component is given as: $\mu_1 = \sum_i C_i$, $\mu_2 = \sum_i (-1)^i C_i$, $\mu_7 = C_1$, $\mu_8 = C_2$, $\mu_9 = C_3$, $\mu_{10} = -C_1 + C_2$, $\mu_{11} = -C_2 + C_3$, $\mu_{12} = -C_3 + C_4$, and the covariance matrix is $(\sigma_{ij})_{(p-4)\times(p-4)}$ where $\sigma_{ij} = 0.5^{|i-j|}$. We generated $A_3$ from a normal distribution with mean $|A_1 + A_2|$ and variance $|A_1|$. $A_4$ has a normal distribution with mean $|A_1 + A_2|^{1/2}$ and variance $|A_2|$. $A_5$ and $A_6$ were generated from Bernoulli distributions with success probabilities $\exp(A_2)/\{1 + \exp(A_2)\}$, and $\Phi(A_2)$, respectively, where $\Phi(.)$ denotes the standard normal cumulative distribution.

**Case 2.** The treatment vector is generated from a multivariate normal distribution where the mean of each component is given as follows. $\mu_1 = \sum_i C_i$, $\mu_2 = \sum_i (-1)^i C_i$, $\mu_3 = C_1 - C_2 - C_3 + C_4$, $\mu_4 = -C_1 + C_2 + C_3 - C_4$, $\mu_5 = \sum_i C_i - 2C_3$, $\mu_6 = \sum_i C_i - 2C_1$, and $\mu_{6+i} = C_i$, $\mu_{9+i} = -C_i$ for $i = 1, 2, 3$, and the covariance matrix is $(\sigma_{ij})_{p\times p}$ where $\sigma_{ij} = 0.5^{|i-j|}$.

The response variable is generated using

$$Y = A^T \beta_1 + (A^T)^2 \beta_2 + \sum_{i=1}^{4} C_i + \Big\{ \sum_{j=1}^{p} A_j \Big\} \times \Big\{ \sum_{i=1}^{4} C_i \Big\} + \epsilon,$$

where the error term $\epsilon$ is generated from standard normal. For $p = 6$, we set $\beta_1 = (1,1,1,1,1,1)^T/\sqrt{6}$, and $\beta_2 = (1,-1,1,-1,1,-1)^T/\sqrt{6}$. For $p = 12$, the last 6 components of $\beta_1$ and $\beta_2$ are identically zero.

As mentioned in Section 3.2, Theorem 1 provides the whole class of estimating equations for a given $\widetilde{U}(\beta)$. For simplicity, we assume $\mathbb{E}[\alpha(A) \mid g(A; \beta)] = 0$, and therefore $\widetilde{U}(\beta) = \{Y - \ell(g(A; \beta))\} \times \alpha(A)$ in the following simulations. The performance of the estimates was computed using the distance between true $\beta$ and $\widehat{\beta}$, defined as the Frobenius norm of the matrix $\widehat{\beta}(\widehat{\beta}^T \widehat{\beta})^{-1}\widehat{\beta}^T - \beta(\beta^T \beta)^{-1}\beta^T$.

**Simulation 1.** In this set of simulations, we aim for evaluating the performance of different estimation strategies for $\beta$ and fix the sample size to 200. The results for both Case 1 and Case 2 when $p = 6$ are presented in Fig. 1, and the results for both Case 1 and Case 2 when $p = 12$ is deferred to the appendix. In each case, there are 4 different boxplots. The first one, from the left hand side, labeled as *Reg*, corresponds to semiparametric SDR estimating equation (1). Since regular SDR ignores the influence of confounding variables $C$, the estimates are not capturing the true causal relationship between $A$ and $Y$. In the second boxplot, labeled as *IPW*, we use the IPW estimator in (7) with the correct model for $p(A|C)$, by properly adjusting for all the confounders. This recovers a more reasonable $\beta^*$ estimate than the first one. However, while *IPW* generally performs better than PCA or regression SDR, the improvement is relatively modest. This might be due to the inefficiency of naive IPW estimators at the reported sample size. The third plot,

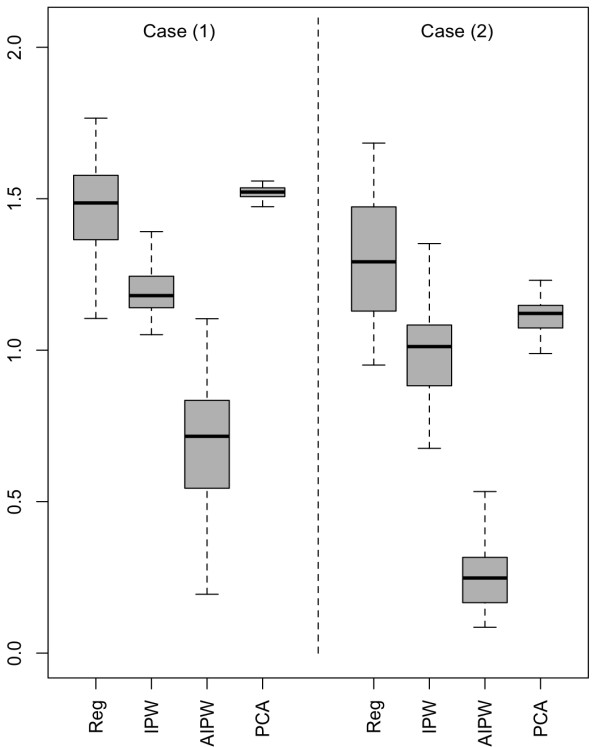

**Boxplots of Euclidean Distances**
**(n = 200, p = 6)**

Figure 1: Boxplots of Frobenius norms between true and estimated parameters in simulations ($p = 6$).

labeled *AIPW*, uses the augmented IPW (AIPW) estimator corresponding to (8), which greatly outperforms the other estimators. The last plot corresponds to the classical PCA dimension reduction technique where the treatment-outcome relation is ignored. In this case, the first two principal directions are reported as estimating the basis of the lower dimensional space. As illustrated in the plots, this naive approach does not seek to preserve a causal, nor indeed *any*, relationship to the outcome.

Our main objective was to reduce the dimension of the treatment such that the cause-effect relation between treatment and outcome is preserved. In order to show that our estimating procedures actually preserve this relation, we compute the contrast between $E[Y(g(a_i; \beta))]$ and $\mathbb{E}[Y(g(a_j; \beta))]$ for $i, j = 1, \ldots, n$, given the true parameters and the estimated ones. The $n \times n$ heatmap of effects are provided in Fig. 2 for the true effects and the ones estimated by regular SDR and AIPW. We used 500 sample points generated from Case 2 with $p = 6$ to plot these heatmaps. The plots in 2(a) and (c) demonstrates the significant similarity between the true surface and the one estimated by AIPW. The surface estimated by regression SDR appears to be a very different surface. The root-mean-squared errors between the true causal surface and the ones estimated from AIPW and

regular regression SDR are 0.48 and 14.29, respectively.

**Simulation 2.** We also evaluate the performance of our bootstrap procedure for estimating the structural dimension $d$, discussed in Section 4. We use the same data generating process as in Simulation 1, with $p = 6$ and $n = 200$. We set the bootstrap size to $B = 50$. The relative frequency of the selected dimension are reported in the appendix, and it reveals that the bootstrap procedure reliably recovers the true structural dimension, namely 2 in both cases (98% of the times in Case 1 and 90% of the times in Case 2.)

**Simulation 3.** We also demonstrate the effect of sample size on *IPW* and *AIPW* estimators of $\beta$ in the causal SDR model. Results are revealed in the appendix.

# 6  DATA APPLICATION

We now illustrate our methods using a cohort of patients treated with radiation therapy for head and neck cancer. The cohort consists of 613 patients who received radiation therapy at the Johns Hopkins hospital prior to 2016. Radiation therapy is one of the most effective modalities for the treatment of head and neck cancers. However, because of the complex shape of target volumes in close proximity to sensitive organs, it may be associated with acute and late radiation morbidities such as xerostomia, mucositis, and dysphagia affecting the patient's quality of life. Such morbidities can lead to severe reduction in food intake and undesirable and possibly dangerous weight loss in patients. There are prospective studies that evaluated risk factors for weight loss in patients who undergo radiation therapy [Johnston et al., 1982, Cacicedo et al., 2014]. However, a proper analysis of whether radiation causes weight loss has not yet been reported likely due to the methodological challenges involved in using high dimensional variables such as radiation therapy as a treatment in causal analysis.

Here, we focus on the parotid glands which are incidentally irradiated by radiation and examine the summary measures of radiation therapy given by the cumulative dose-volume histograms extracted from the raw voxel maps of radiation doses. In particular, we looked at 5 equally spaced percentages of volume to construct a vector of treatment doses. We used weight loss as the outcome of interest, which was defined as the difference between weight measured within 100 to 160 days after the completion of treatment and the weight measured during consultation before the start of treatment. The data has records on demographics such as age, gender, race, and baseline clinical factors such as whether the patient had used feeding tubes and/or received chemotherapy before the initiation of treatment. We assumed these variables are sufficient to control for confounding and thus would ensure the conditional ignorability assumption was met. A copy of this dataset is available on https://github.com/raziehna/multidimensional-treatments.

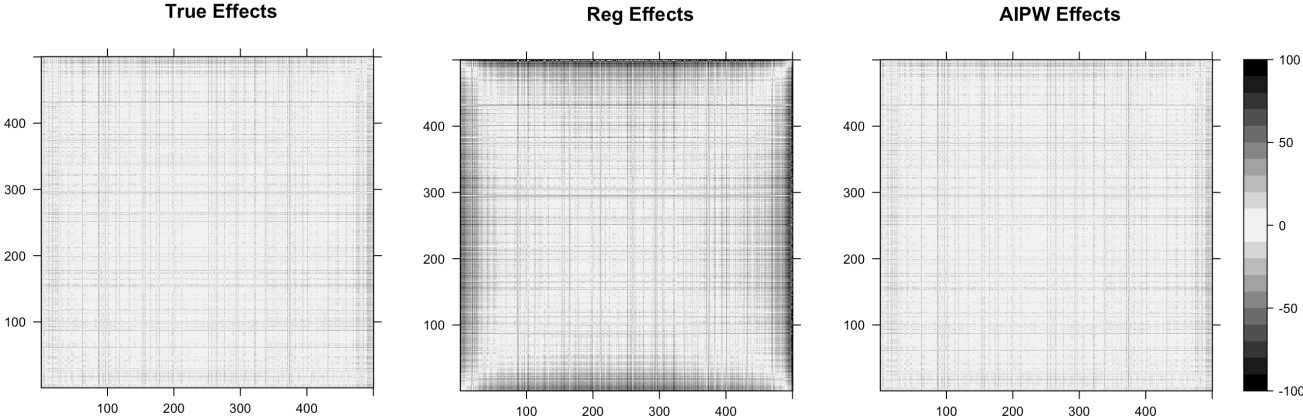

Figure 2: Heatmaps of true causal effects and effects computed by estimating $\beta$ via the regular SDR and the AIPW estimators. Heatmaps are antidiagonally symmetric.

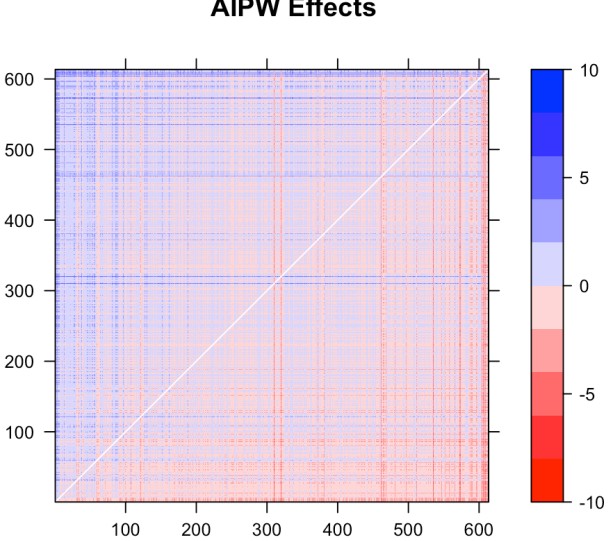

Figure 3: Heatmap to illustrate the causal effect of radiation on weight loss, where effects are computed by estimating $\beta$ via AIPW estimator. Heatmap is antidiagonally symmetric with opposite color tones.

There exists a rich literature relating parotid dose-volume characteristics to radiotherapy-induced salivary toxicity. It has been shown that the mean dose to the parotid glands correlates strongly with xerostomia and salivary dysfunction which are risk factors of weight loss [Deasy et al., 2010]. In light of such studies, we assume there exists a single dimension in the radiation exposure that captures the relationships between exposure and side effects including weight loss. Therefore, we set the structural dimension $d$ to be one. We set the mapping function $g(.; \beta)$ to be linear in its parameters $\beta$, and use Bayesian additive regression trees to fit all nuisance models. The code is provided as part of the supplementary materials. The oncology data were excluded for reasons of patient confidentiality.

We generated $n \times n$ heatmaps in Fig. 3 to illustrate the cause-effect relationship between radiation treatment and weight loss. We use AIPW estimator obtained from Theorem 1. The absolute values on the plots are antidiagonally symmetric. Radiation doses were sorted in increasing values along both axes. We interpret the heatmaps as follows. Consider the $(i, i)^{\text{th}}$ point on the plot and draw a line along the y-coordinate. Since radiation doses were sorted in increasing order, then the radiation value at any point on the line to the right of $(i, i)$ is higher than the radiation value at the $(i, i)^{\text{th}}$ point. For any point to the left of $(i, i)$, the radiation value is lower. The value at the $(k, i)^{\text{th}}$ coordinate corresponds to the contrast $\mathbb{E}\big[Y(g(a_k; \beta)) - Y(g(a_i; \beta))\big]$. Consequently, if $k > i$, then a red dot at $(k, i)$ coordinate implies that an increase in radiation doses leads to an increase in weight loss. On the other hand, a blue dot would imply that an increase in radiation doses would not lead to an increase in weight loss. Similarly, a blue dot at $(k, i)$, for $k < i$, would imply that a decrease in radiation leads to a decrease in weight loss. Reverse is implied when the dot is red. Focusing on the bottom right triangle, we note that most of the area is filled with red color. It implies that as we increase the amount of radiation, the severity of weight loss increases. Thus, radiation therapy is potentially a cause of weight loss among patients who undergo the treatment.

We investigated the relationship between the treatment and outcome as the treatment size increases by selecting larger numbers of equally spaced percentages of volume in the dose-volume histograms. The plots are provided in the supplement. Throughout the experiment, we examined the summary measures of radiation therapy given by the cumulative dose-volume histograms extracted from the raw 3D voxel maps of radiation doses. A more fine-tuned approach is to look at the exact dose localization information in the raw 3D voxel maps. A voxel-based approach would identify the relations between radiation-induced morbidity and local dose release, thus providing a potentially better insight

into spatial signature of radiation sensitivity in composite regions like the head and neck district [Monti et al., 2017]. Given the small cohort of patients that we have access to, a voxel-based approach would fall into $p \gg n$ paradigm, and would require strong sparsity assumptions [Li, 2007] to deal with. This is an interesting and challenging direction for future work.

# 7 CONCLUSIONS

In this paper, we have described a generalization of the semi-parametric sufficient dimension reduction (SDR) approach for regression problems described in [Ma and Zhu, 2012] to causal SDR. Specifically, we developed a method that reduces the dimension of a multidimensional treatment, while preserving the causal relationship between the treatment and the outcome quantified as a counterfactual mean. Using ideas from structural models [Robins, 1999], we provided semiparametric estimators for parameters of the function that maps the multidimensional treatment to a lower dimensional subspace. We have shown our estimator exhibits "2x2 robustness," where the estimator remains consistent if one of two models, for two pairs of models, is chosen correctly.

Even though we use radiation therapy as our main motivation example, our methodology can be used in other applications as well. For instance, in natural language processing, a growing literature is focused on understanding causal effects of high dimensional text data; see [Feder et al., 2021]. Another example is relating neuronal network activity (collected via high dimensional neuroimaging data) to cognitive processing and behavior; see [Ramsey et al., 2010, Mather et al., 2013] for examples. Our proposed framework can also be useful in settings where we are interested in causal effects of multiple treatments simultaneously. In order to scale our methods to high dimensional applied settings, such as fMRI scans, text data, or radiation oncology voxel data, we need to incorporate ideas from parametric modeling, and sparsity within a semiparametric framework. Another natural extension for future work is to apply these methods to classical causal inference in longitudinal studies, where multiple time points render a collection of binary treatments a multidimensional object. Our causal SDR approach would provide an alternative to parametric marginal structural models typically employed in such settings.

## Author Contributions

RN and IS contributed to conception and design of the statistical framework. RN performed the statistical analysis. TM provided the radiation database, and was the key in formulating the challenge of multidimensional treatments in oncology and radiation therapy. All authors contributed to the write up and revision of the manuscript.

## Acknowledgements

We thank Todd McNutt's research group for their insightful discussions and facilitating our access to the database on head and neck cancer patients. We thank the anonymous reviewers for their comments. Ilya Shpitser is sponsored in part by NSF CAREER: 1942239, ONR N00014-21-1-2820, NIH R01 AI127271-01A1, and NSF 2040804.

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
