# OpenReview forum: "Semiparametric Causal Sufficient Dimension Reduction of Multidimensional Treatments"
_auai.org/UAI/2022/Conference — UAI 2022 Poster_

### Official Review · Reviewer_bjD5 · 2022-04-09

**Q2(1) Originality/Novelty:** 2
**Q2(2) Significance/Impact:** 2
**Q2(3) Correctness/Technical Quality:** 2
**Q2(6) Clarity Of Writing:** 1
**Q6 Overall Score:** 3
**Q8 Confidence In Your Score:** 2

**Q1 Summary And Contributions:**

The paper studies the problem of causal effect estimation of multidimensional exposures, such in the case of radiation therapy. It focuses on reducing the dimensionality of exposure space,  while keeping causal relations between exposures and outcomes.  It adapts the approach of “sufficient dimension reduction” for preserving causal relations using either IPW or AIPW.

**Q2 Assessment Of The Paper:**

More detailed information regarding each of these aspects is given below:

**Q2(4) Quality Of Experiments (Optional):**

2: Fair: The experimental evaluation is weak: important baselines are missing, or the results do not adequately support the main claims.

**Q2(5) Reproducibility:**

1: Poor: Key details (e.g., proof sketches, experimental setup) are incomplete/unclear, or key resources (e.g., proofs, code, data) are unavailable.

**Q3 Main Strengths:**

1.	Reducing the dimension of multi-dimensional interventions is of interest and practical importance, e.g., for explainability.
2.	The proposed method (SDR with AIPW) was novel to me.
3.	The paper present results over simulated data, where ground truth is known, showing the superiority of the proposed method over 2 (non-causal) dimensionality reduction methods: PCA and SDR.


**Q4 Main Weakness:**

1.	The paper is very hard to read and is not self-contained. It uses much terminology that is not explained in the main body of the paper, such as: “influence functions”, “nuisance tangent space”.  I failed to follow the methodology and theoretical results in the paper.  I provide more details in Q5.
2.	I did not see any mention of other studies of causal effect estimation with multi-dimensional exposures (e.g. text: Egami, Naoki, et al. "How to make causal inferences using texts."). In particular, I would like to see a comparison to outcome-based methods (“g-formula”) without dimensionality reduction, e.g. BART.


**Q5 Detailed Comments To The Authors:**

I list here several of the many points in the paper I failed to understand:
1.	Can you shortly describe “influence functions”, and not just refer the reader to the appendix and other papers?
2.	What is the role of \alpha(X)? it says “\alpha(X) is any function of X”, but what does it represent? Does it represent the propensity function we try to estimate?
3.	What is \eta_a ? I did not see any definition of it.
4.	In the Pparagraph following Equation 4: “This IPW procedure is known to be inefficient”. Can you explain why?
5.	What does q(Y, A, C) and p*(A) represent? Is p* (A) an estimation for p(A)?
6.	Can you define what are “regular and asymptotically linear (RAL) estimators”?
7.	What is a “normalized function”? does it mean a density function (i.e. integral = 1?)
8.	The procedure described under “implementation” is very hard to follow. It would be beneficial to have a simplified, self-contained procedure that one can implement without having to go over previous theoretical results.


**Q7 Justification For Your Score:**

In its current form, the paper is completely non-readable for people not from the field. I am familiar with the basics of causal inference, e.g., the ones presented in the book “Causal Inference” by Hernan & Robins, and read many papers on causal effect estimation, but I couldn’t follow this paper.

**Q9 Complying With Reviewing Instructions:**

1: Yes.

---

### Official Review · Reviewer_MKBP · 2022-04-11

**Q2(1) Originality/Novelty:** 3
**Q2(2) Significance/Impact:** 3
**Q2(3) Correctness/Technical Quality:** 3
**Q2(6) Clarity Of Writing:** 3
**Q6 Overall Score:** 7
**Q8 Confidence In Your Score:** 4

**Q1 Summary And Contributions:**

The paper proposes a semiparametric causal sufficient dimension reduction approach, which reduces the dimension of a multidimensional treatment, while preserving the causal relationship between the treatment and outcome. The paper also provides theoretical properties for the proposed estimator and detailed implementation algorithm.


**Q10 Ethical Concerns (Optional):**

There is no ethical concern.

**Q2 Assessment Of The Paper:**

More detailed information regarding each of these aspects is given below:

**Q2(4) Quality Of Experiments (Optional):**

3: Good: The experimental evaluation is adequate, and the results convincingly support the main claims.

**Q2(5) Reproducibility:**

3: Good: Key resources (e.g., proofs, code, data) are available and key details (e.g., proofs, experimental setup) are sufficiently well-described for competent researchers to confidently reproduce the main results.

**Q3 Main Strengths:**

The introduction of the new approach is detailed and clear.

There is some theoretical support for the proposed estimator.

Simulation study shows that the proposed AIPW method has better performance compared with several other methods.

Real data analysis shows how to apply the proposed method to practical problems.

**Q4 Main Weakness:**

There are some minor errors in the paper.

**Q5 Detailed Comments To The Authors:**

In the first paragraph of Section 4, it says that "The last term in (8) is equal to ...". I think here U should be \tilde{U}.

The Step 5 of the implementation procedure may have some problem, please check it carefully. Can you explain more about how to calculate the first and second derivatives numerically?

In the simulation study, what is the form of \alpha(A)?

In the simulation study, can you show some simulation results to verify that the proposed AIPW method has the robustness properties?

There are some typo errors in the paper, please check it carefully. For example, on Page 5, step (c) should be changed to step 3.

**Q7 Justification For Your Score:**

I make this score based on the main strengths,  weaknesses and my understanding of this paper. I think the strengths of this paper far outweigh its weaknesses.

**Q9 Complying With Reviewing Instructions:**

1: Yes.

---

### Official Review · Reviewer_1cBL · 2022-04-11

**Q2(1) Originality/Novelty:** 3
**Q2(2) Significance/Impact:** 3
**Q2(3) Correctness/Technical Quality:** 3
**Q2(6) Clarity Of Writing:** 2
**Q6 Overall Score:** 4
**Q8 Confidence In Your Score:** 3

**Q1 Summary And Contributions:**

The authors show how causal inference may be improved by skillfully reducing the dimension of multidimensional data without worsening inference, using semiparametric inference theory.

**Q2 Assessment Of The Paper:**

More detailed information regarding each of these aspects is given below:

**Q2(4) Quality Of Experiments (Optional):**

4: Excellent: The experimental evaluation is comprehensive and the results are compelling.

**Q2(5) Reproducibility:**

3: Good: Key resources (e.g., proofs, code, data) are available and key details (e.g., proofs, experimental setup) are sufficiently well-described for competent researchers to confidently reproduce the main results.

**Q3 Main Strengths:**

Previous work is extended in a useful direction by means of a semiparametric algorithm for reducing dimension. The approach is well-situated in the literature.

**Q4 Main Weakness:**

Although the results here can't be reproduced without the oncology dataset used to generate them, it is understandable that this data could not be included for reasons of confidentiality, although perhaps a different dataset that could be publicly shared might have been a better choice given UAI's policy for including data.

Also, I will take the authors' word for it that code is included in supplementary materials, although I don't seem to have received that as part of the review package. Nevertheless, there is a promise to make that public.

After review--I do see the code now and the simulation example, so that much can be reproduced. Thanks.



**Q5 Detailed Comments To The Authors:**

I found this paper to be a bit hard to follow. There are a number of oblique references to other work that were not explained in the context of this text, and I didn't have time to track them all down to help me follow every paragraph. I apologize for this.

**Q7 Justification For Your Score:**

I thought this was a good paper, but I did have trouble following some of the details even after reading it a few times. The algorithm is good, and the results are good. Also, I was pretty sure an arbitrary expert couldn't reproduce the results, since they wouldn't have access to the oncology dataset that was used.

**Q9 Complying With Reviewing Instructions:**

1: Yes.

---

### Official Review · Reviewer_EdTw · 2022-04-13

**Q2(1) Originality/Novelty:** 2
**Q2(2) Significance/Impact:** 2
**Q2(3) Correctness/Technical Quality:** 3
**Q2(6) Clarity Of Writing:** 4
**Q6 Overall Score:** 6
**Q8 Confidence In Your Score:** 3

**Q1 Summary And Contributions:**

The paper deals with the problem of reducing the dimension of the treatment variable under preservation of causal relationships to the outcome variables. The goal is to better understand and interpret causal effects even in settings with high-dimensional treatments.

**Q2 Assessment Of The Paper:**

More detailed information regarding each of these aspects is given below:

**Q2(4) Quality Of Experiments (Optional):**

3: Good: The experimental evaluation is adequate, and the results convincingly support the main claims.

**Q2(5) Reproducibility:**

3: Good: Key resources (e.g., proofs, code, data) are available and key details (e.g., proofs, experimental setup) are sufficiently well-described for competent researchers to confidently reproduce the main results.

**Q3 Main Strengths:**

The paper deals with the novel setting of reducing the dimension of the treatment, in order to obtain interpretable causal effect estimates. This is done by adapting SDR (sufficient dimension reduction) ideas towards the causal setting.

The paper is clear and well-written. The merits of the approach are shown in simulations and on real data.

**Q4 Main Weakness:**

I like the paper in general. The setting is rather specific (compared to the more common dimension reduction of confounding variables), but in my view has been sufficiently motivated. On the technical side, the paper is building on the causality and SDR literature and combines ideas from these fields, but the results are certainly not trivial.

**Q5 Detailed Comments To The Authors:**

- in the experiments on real data you write that a crude summary of the treatment has been used (with dim > 1). Could you expand on this a bit more/give some details in the appendix?
- Writing:
  - the secondd paragraph of the main text lacks a point at the end
  - page 3: and maps them to values

**Q7 Justification For Your Score:**

The problem dealt with in this paper is quite specific, but in my view sufficiently motivated. Hence, I weigh the strengths more heavily.

**Q9 Complying With Reviewing Instructions:**

1: Yes.

---

### Decision · Program_Chairs · 2022-05-15

**Decision:**

Accept (Poster)

**Comment:**

Meta Review: This is a well-written paper offering a formal semiparametric procedure to estimate the average causal effect of a dimensionality reduced representation of the cause. The paper develops the theory and applies it to simulated and real data. The code has been made available and the authors have promised to try to make an anonymized version of the real data available as well.
I consider my recommendation to be in line with the reviewers. Below are points both from the reviewers and from my own reading of the paper.

pros
-- code and data are or are going to be made available
-- development of theory for the estimation of causal effects of aggregate quantities is important and of interest;
-- the authors made a substantive effort to address the reviewers' questions and the reviewers felt largely satisfied with those responses and are confident that the authors will improve the paper accordingly

cons
-- the paper is clearly written but comes from a background in semiparametric theory in statistics. This makes it less accessible to an ML audience, and connections to work on causal representation learning in ML or knock-off testing in statistics are missed. Several reviewers noted that this paper was difficult to read without substantial background in semi parametric inference. I think the paper is well done but would strongly encourage the authors to not dismiss the concerns raised about accessibility of the paper. They are unnecessarily limiting their impact. The problem that is addressed in the paper has been addressed in several variations in ML and stats outside of semi parametric theory. The detailed assumptions and results are slightly different in each case, but the overall aims and concerns are the same. See, e.g. work by Chalupka et al on Causal Feature Learning, by Kun Zhang et al on the GIN condition for learning latent variable causal models, the different approaches using knock-off tests for GWAS studies that cluster the genes (Matteo Sesia has an online tutorial) or the approaches using auto encoders that Caroline Uhler's group uses to learn a latent representation and the causal relation. In other domains, Canonical Correlation Analysis is used for this sort of problem in the linear setting. This list is not exhaustive, but should indicate that comparisons to simple PCA, as done here, are perhaps a bit limited. (Of course there is a page limit, but I did not get a sense that there was an awareness of this broader space of approaches.)
-- interpretability: this is not a point that was discussed in the reviewing cycle, so I want to bring it up here: The authors claim that the procedure improves interpretability of the causal conclusions. But this claim is not spelled out. In what sense are the results clearly or better interpretable? As far as I can tell, the interpretation of the single dimension (that had been fixed a priori) in the actual data application was derived from background knowledge (values in the plots were sorted accordingly), not inferred from the output of the method. In the intro the authors suggest that one may be interested in the entire dose-response relationship --- that sounds like the reduced dimension is pre-set, not learned. More generally, what is the appropriate interpretation of an aggregate cause when its parts are prone to not have a significant causal effect (as suggested in the intro) and the aggregate is sensitive to the mixing distribution P(A|C) over the cause?
-- unless I am missing something, the simulations are entirely linear Gaussian, even though the theory is broader.